# High Definition Three-Dimensional Exoscope (VITOM 3D) in E.N.T. Surgery: A Systematic Review of Current Experience

**DOI:** 10.3390/jcm11133639

**Published:** 2022-06-23

**Authors:** Salvatore Ferlito, Ignazio La Mantia, Sebastiano Caruso, Giovanni Cammaroto, Carlos Miguel Chiesa-Estomba, Giannicola Iannella, Francesco Nocera, Angelo Ingrassia, Salvatore Cocuzza, Claudio Vicini, Stephane Hans, Jerome Rene Lechien, Antonino Maniaci

**Affiliations:** 1Department of Medical and Surgical Sciences and Advanced Technologies “GF Ingrassia” E.N.T. Section, University of Catania, 95123 Catania, Italy; ferlito@unict.it (S.F.); igolama@gmail.com (I.L.M.); sebastiano.caruso2404@gmail.com (S.C.); ciccionocera94@gmail.com (F.N.); angeloingra94@gmail.com (A.I.); s.cocuzza@unict.it (S.C.); 2Department of Otolaryngology-Head and Neck Surgery, Morgagni Pierantoni Hospital, 47121 Forli, Italy; giovanni.cammaroto@hotmail.com (G.C.); claudio@claudiovicini.com (C.V.); 3Department of Otorhinolaryngology-Head and Neck Surgery, Hospital Universitario Donostia, 20014 San Sebastian, Spain; chiesaestomba86@gmail.com; 4Otorhinolaryngology Department, Sapienza University of Rome, Policlinico Umberto I, Viale del Policlinico, 00185 Rome, Italy; giannicola.iannella@uniroma1.it; 5Department of Otorhinolaryngology and Head and Neck Surgery, Foch Hospital, School of Medicine, U.F.R. Simone Veil, Université Versailles Saint-Quentin-en-Yvelines (Paris Saclay University), 92150 Paris, France; prhans.foch@gmail.com (S.H.); jerome.lechien@umons.ac.be (J.R.L.); 6Department of Human Anatomy and Experimental Oncology, Faculty of Medicine, UMONS Research Institute for Health Sciences and Technology, University of Mons (UMons), 3800 Mons, Belgium; 7Department of Otorhinolaryngology and Head and Neck Surgery, CHU de Bruxelles, CHU Saint-Pierre, School of Medicine, Université Libre de Bruxelles, 1060 Brussels, Belgium

**Keywords:** 3D, exoscope, E.N.T. micro-surgery, microscope, head and neck oncology

## Abstract

Over the last decade, technological growth has developed new devices for more precise surgery based on improved maneuverability, minimally invasive approaches, and magnification of the operating field. In this context, the exoscope has opened a new phase for more accurate and safer microsurgery, improving the perception of the volume of objects and the depth of structures for planning, targeting, and controlling fine movements. The exoscope could be used for middle ear, transcanal, transmastoid, and craniotomy procedures that require two-handed dissection, both to perform both totally VITOM-based techniques and coupled to traditional procedures with an operating microscope or endoscope. In addition, the VITOM 3D system allows the surgeon to work with high-definition images, which is essential in facial nerve surgery or submandibular salivary stone or tear surgery approaches, where magnification plays a fundamental role in surgical success and in reducing operating times. The 3D exoscope approach could also be included in traditional transoral procedures for oropharyngeal carcinoma. The exoscope may provide a relevant approach in teaching surgeons and nurses, allowing adequate training in non-oncological surgical procedures such as a tonsillectomy or lateral pharyngoplasty.

## 1. Introduction

During the last decade, the exoscope became an addition to the microsurgical armamentarium, which was designed to replace the operative microscope. The exoscope consists of a high-definition or 4K video camera with optical and digital zoom and a fiber optic-delivered or L.E.D. light source [1,2]. This system is suspended above the surgical field with a manually actuated articulating holder or robotic arm, which transmits a two-dimensional (2D) or 3D image to a high-resolution monitor placed at eye level directly across from the surgeon. Magnification power of 8–30 is possible through the 3D camera, while the depth of field ranges from 7 to 44 mm, with a focal distance of 20–50 cm, allowing the surgical field to be observed and illuminated at various distances from the patient [2,3,4,5]. The 3D exoscope provides surgeons with a feasible and potentially excellent alternative to traditional operating microscopes and endoscopes in head and neck surgery. In contrast to endoscopes, the 3D exoscope has distinct advantages: the depth of structures, the perception of objects’ volume, a longer focal distance (creating ample working space), and the ability to easily adjust the surgical view without anatomic constraints. The exoscope uses an external display, allowing the surgeon to have a more comfortable posture, sharing the surgical field of view [2,3,6,7,8].

## 2. Materials and Methods

### 2.1. Protocol Data Extraction and Outcomes

Three authors analyzed the data from the literature (A.M., S.C., and F.N.). Any disagreements were resolved through discussion by our study team. The included studies were then analyzed to gather all available data and guarantee eligibility among enrolled subjects. In addition, the patient’s diseases, diagnostic procedures, treatment modalities, and main polymorphisms were collected. The following information was collected: author data, year, sample size (4k exoscope-assisted group vs. traditional surgery), study design, statistical analysis, findings, and conclusions. We contacted the authors of the included studies if the required data were not complete, using the corresponding author’s email or ResearchGate (http://www.researchgate.net/, accessed on 15 March 2022). 

### 2.2. Electronic Database Search

Following the PRISMA checklist for review and meta-analysis, a systematic review of the current literature and the PICOS search approach was performed [9,10]. The research protocol was submitted and recorded on the PROSPERO database (code 300827).

We searched PubMed, Scopus, and Web of Science electronic databases for studies on 4k exoscope-assisted surgery patients suffering from E.N.T. disorders over 20 years (from 1 January 2001 to 1 December 2021) by three different authors, using MeSH, Entry Terms, and related keywords. The related search keywords used were as follows: “3D exoscope surgery; exoscope 4K assisted surgery”, “E.N.T. Microsurgery”, “VITOM assisted surgery”, and “Head and Neck 3D surgery”.

The ‘Related articles’ option on the PubMed homepage was also considered. We used reference manager software (EndNote X7^®^, Thomson Reuters, New York, NY, USA) to collect references and remove duplicates. The titles and abstracts of papers available in English were examined by the investigators (A.M., S.C., and F.N.). The identified full texts were then screened for original data, and the related references were retrieved and checked manually to identify other relevant studies.

### 2.3. Eligibility Criteria

The PICOS approach was adopted to assess eligibility, including Medical Subject Headings (MeSH), Entry Terms, and keywords found in articles in this field. The following items were considered: Participants (E.N.T. patients); Intervention (exoscope-assisted surgery, other traditional surgery); Control (applied); Outcome (surgical success, fewer peri or postoperative complications, better control of intraoperative pathologies); Study type (observational study). 

We imposed language, publication date, and publication status as restrictions. Thus, we considered surgical success of E.N.T. disorders treated with 4k exoscope-assisted surgery as the primary outcome. Other parameters assessed in the studies were considered secondary outcomes.

We included all studies that met the following criteria:Original articles;The article was published in English;The studies included clinically confirmed E.N.T. disorders treated with 4k exoscope-assisted surgery;The studies reported detailed information on 4k exoscope-assisted surgery as high-definition images of the surgical field, improved vision and depth perception, different treatment modalities, and patient’s comorbidities.

The exclusion criteria were as follows:5.Editorials, letters to the editor, or reviews;6.Studies that included animal samples.

### 2.4. Synthesis of Results

Given different E.N.T. disorders, surgical procedures performed, tissue samples, and the follow-up programs adopted, quantitative analysis was not performed because of the influence on the outcomes reported. Consequently, we performed a narrative synthesis following the synthesis guidelines, without meta-analysis reporting items [11].

### 2.5. Statistical Analysis

The research protocol was carried out according to the approved reporting items’ quality requirements for systematic review and meta-analysis protocol (PRISMA) declaration [9]. We adopted the studies’ quality assessment (QUADAS-2) instrument to estimate the included studies’ design features, and the results for the risk of bias are presented descriptively [12]. Moreover, the potential risk of bias in observational studies was assessed using the Joanna Briggs Institute Critical Assessment Checklist for Observational Studies [13].

## 3. Results

### 3.1. Paper Retrieval 

The systematic review of the literature identified 239 potentially relevant studies (Figure 1). After removing duplicates and applying the criteria listed above, 187 records were potentially relevant to the topic. Through the analysis of records and subsequent full-text screening of the articles, we excluded all the studies that did not match the inclusion criteria (n = 156). The remaining 22 papers were included in a qualitative synthesis for data extraction [1,2,4,5,14,15,16,17,18,19,20,21,22,23,24,25,26,27,28,29,30,31]. Moreover, according to the criteria established for meta-analysis, a quantitative analysis was not performed. A graphical display of bias analysis outcomes is shown in Figure 2, summarizing the possible risk of bias. 

### 3.2. Study Features

We included 22 studies in the analysis [1,2,4,5,14,15,16,17,18,19,20,21,22,23,24,25,26,27,28,29,30,31]. According to the study design, we identified 5 papers as case reports [16,20,22,29,31], 2 papers as case series [15,24], 10 papers as prospective controlled studies [2,4,5,14,17,23,25,27,28,30], and 5 studies as retrospective studies [1,18,19,21,26]. The studies’ sample sizes ranged from 1 to 71 participants. A total of 303 participants were assessed. The relevant data retrieved from the included original studies are described in Table 1. 

The quality of evidence evaluation conducted by the GRADE assessment was considered low. This was mainly because of the study design (observational studies), heterogeneous methodology, and risk of bias in the included studies. The evidence appraisals are summarized in Figure 2.

### 3.3. Patients’ Features, Comorbidities, and Treatment

The patients’ average age was 54 ± 16.41 years, ranging from 11 to 82 years old. The major surgical procedures performed were otomastoid and skull base surgery in 101 cases (33.33%) [1,14,15,16,17,18,19], salivary gland surgery in 95 cases (31.35%) [4,26,27,28,29], nasal and paranasal surgery in 31 cases (10.23%) [19,20,21,22]. In addition, laryngeal surgery was executed with an exoscopic approach in 28 cases (9.24%) [2,23,24]. Head and neck surgery was performed in 38 cases (12.54%) [5,25,30,31] (Figure 3). 

### 3.4. Surgical Times and Cost Effectiveness in Comparision of Techniques

The application of the VITOM system to surgical procedures has not shown a significant increase in surgical time compared to traditional methods [1,18]. Rubini et al., during lateral skull base surgery procedures, reported a significantly shorter mean operative time in the exoscope group than the microscope group (289 vs. 313 min; *p* < 0.05) [18]. A close correlation of shortened surgical times was also a cost-effective benefit of the exoscope over a modern surgical microscope. Crosetti et al. (2020) reported a low consumables cost (VITOM and surgical coating) for each procedure of €62 (€41 and €21, respectively), while the effectiveness between TORS and microscope approaches was comparable [5].

Bignami et al. (2021) reported an easy frontal fibro-osseous removal thanks to the easy transition from the assistant role to the first operator, with consequent advantages in terms of cost effectiveness [22]. However, it is reported that the surgical time and maneuverability of the system could be limited if a deeper surgical field is required, in the presence of poor lighting in small surgical corridors and when high magnifications are required [15,17,18,23]. Minoda et al. found higher magnification in middle ear cholesteatoma surgery; however, there could be a deterioration of the surgical images, with consequent pixelation [15]. Excessive visual fatigue due to the use of polarizing glasses for the 3D view for the duration of the procedure has also been linked to headaches and dizziness [25,26,31].

### 3.5. Lateral Skull Base and Ear Surgery

Among the different techniques of otological surgery, post-aural approaches appear to be the most suitable for exoscopic surgery; therefore, the VITOM 3D system can be used in the treatment of chronic pathologies of the middle ear and several tumor types of the external and middle ear, and to perform cochlear implants [1,14,15,16,17,18].

In 2018, Smith et al. stated that the 3D exoscope advantages could present a valid choice over the operating microscope for neurotological surgery [17]. The authors stressed that, unlike the microscope, the exoscope allowed better communication and the exchange of instruments between the members of the surgical team, thanks to an optimal display on the monitor and the easy maneuverability of the instrument. Although the binocular microscope has long been considered the primary visualization tool for neurotological and skull base surgery, it forces otolaryngologists to assume uncomfortable positions with little freedom of movement, resulting in an increase in muscle complications and skeletal features of the operators.

Consequently, in 2019, alternative middle ear cholesteatoma surgery was described by Minoda et al., including two cases of mastoid cavity involvement [15]. The authors evaluated the feasibility of a retro-auricular transcortical mastoidectomy using a 3D surgical exoscope, while the consequent cholesteatoma removal was performed through the mastoidectomy opening with an endoscope. It was described as head-up surgery, performed watching a monitor, and was ergonomically more suitable, allowing operation in a physiologically comfortable position. The authors did not find any deterioration of the postoperative bone conduction data and residual cholesteatoma during the second-stage surgery at 9 months.

In 2021, Wierzbicka et al. described their initial experience with the high-definition three-dimensional exoscope for middle ear surgery by comparing it with the operating microscope [14]. The research enrolled 60 selected patients diagnosed with otosclerosis (n = 30) or chronic otitis media (n = 30) and indicated for surgery. The primary purpose was to evaluate the quality of the visibility of the operating field provided by the VITOM-3D exoscope compared to the operating microscope. The tympanoplasty procedure was completed with the 3D exoscope in 28/30 cases (93.3%), while the conversion to the operating microscope was performed in 2/30 cases (3.3%) because of bleeding during removal of cholesteatoma and granulation from the tympanic cavity. In both stapedotomy and tympanoplasty, the exoscope was superior to the microscope during the more superficial parts of the procedures; however, in the deeper areas of the middle ear, the exoscope was critical. In fact, both intraoperative bleeding and the restricted surgical field reduced its visibility. The study shows that the VITOM 3D provides excellent high-definition images of the surgical field but has several important limitations, such as reduced depth perception in deep areas of the tympanic cavity and reduced visibility in a difficult surgical field, forcing the surgeon to choose an operating microscope in selected cases [14].

Larger studies with a more systematic evaluation of the 3D exoscope image quality, ergonomics, and impact on surgical eye fatigue and workflow could aid in evaluating the usefulness of the exoscope in E.N.T. surgery [17]. 

In 2019, Smith et al. performed 11 otological procedures with the VITOM 3D system together with or in place of the operating microscope, including cochlear implant surgery, two resections of the vestibular schwannoma eventually coupled with the operative microscope, and visualizing the tumor from the cerebellopontine angle [17]. Instead, seven patients were treated with an exoscope alone, with a subjective reduction in neck fatigue with respect to an operative microscope (*p* = 0.03) and no difference in manipulation or visualization of structures (*p* = 0.05).

Recently, Colombo et al. evaluated the potential of the 3D exoscope in specific ear procedures; the innovative technology was retrospectively compared with surgeries treated with an operating microscope [1]. The authors enrolled 13 patients for each group, including 9 tympanoplasties, 1 acute complicated mastoidectomy, 1 revision myringoplasty, and 2 cochlear implants. No statistical differences were found among the procedures in operating room time; the exoscope was lighter, had better maneuverability, and compactness, while the need for an extensive surgical corridor and the rendering of bright structures were the main limits.

Further studies, including microscope comparison and cost-effectiveness investigations, with more representative samples and with real clinical scenarios, are necessary to understand its real potential in clinical practice [14]. Rubini et al. (2019) published a retrospective study that included 24 patients affected by lateral skull base pathologies who underwent surgery using the 3D exoscope or the operative microscope at the Department of Otolaryngology—Head and Neck Surgery at the University Hospital of Verona, Italy [18]. The exoscope and microscope groups each included 12 cases. The feasibility of all the surgical steps solely using the 3D exoscope was evaluated. 

Surgical time, facial and hearing function outcomes, and intraoperative or postoperative complications were analyzed, demonstrating no intraoperative complications during all the procedures. In contrast, at follow-up, one complication occurred. The authors reported an average surgical time of 289 min in the exoscope group and 313 min in the microscope group, not revealing statistical differences (*p* > 0.05). Moreover, the facial and hearing function outcomes were similar [18].

### 3.6. Nasal and Paranasal Surgery

Nasal and paranasal surgery could benefit from exoscopic and 3D techniques, as the outstanding visualization of the anatomical structures is an essential element for surgical success [19,20]. Optical magnification plays an essential role in rhinology techniques, particularly after the spread of endoscopic approaches for nasal surgery. In 2016, Tasca et al. described the use of VITOM 3D technology in rhinoplasty procedures [20]. The authors reported a dramatic improvement in visualization of the surgical field, understanding of procedures, and the teaching environment. The supporting device was rotated in the three space planes to follow the flow of the surgery. In this way, the operating field may always be centered on the screen, even in cases of inevitable movements of anatomical structures during operating maneuvers such as elevations of the tunnels or osteotomies.

An interesting study was carried out by Pirola et al. [21], who proposed the application of the VITOM 3D exoscope in a group of 21 patients undergoing mono or bilateral dacryocystorhinostomy (DCR). The exoscope usage was evaluated by a team of expert surgeons possessing consolidated expertise derived from >400 cases of DCR performed in the previous 10 years of practice. The introduction of the exoscope in DCR was “completely approved” in 55.5% of cases, “moderately approved” in 39.7%, and “weakly approved” in 4.8%. Their first concern was to verify the non-inferiority of the combined exo-endoscopic approach to the classic setting for DCR. Indeed, patients’ outcomes at the 6-month follow-up and time for surgery were identical between the two groups [18]. Moreover, no effects on symptom outcomes (such as epiphora or dacryocystitis rates) and mean surgical time compared to classic DCR were detected.

Recently, Bignami et al. reported the first experience of frontal fibro-osseous lesion removal, associating a 3D-4K exoscope to a coupled system with a dedicated robotic arm (ARTip Cruise ™) via a combined osteoplastic flap-endonasal approach [22]. The patient reported a 6-month history of recurrent frontal sinusitis accompanied by left frontal swelling and epiphora. The surgical procedure did not require operative microscope usage or direct vision, consenting a near-total resection, and the patient remained asymptomatic after 1 month.

### 3.7. Exoscope Application in Head and Neck Surgery

#### 3.7.1. Oral Surgery 

The surgical approach to squamous cell carcinoma of the oral cavity and oropharynx plays a fundamental role in the management of the disease, both in the presence of early and advanced disease stages. However, the different techniques developed have always suffered from poor accessibility and visualization of the anatomical structures of the oral cavity, forcing surgeons to choose invasive approaches for the patient. The term 3D exoscopic surgery was coined to describe the technique used for transoral surgery with the exoscopic system as a visual tool [5]. Crosetti et al. (2018) reported 10 patients with oropharyngeal carcinoma treated with oropharyngeal surgery with a 3D exoscopic surgery approach. In all patients, the VITOM 3D system has proven to be a versatile and compact optical instrument that provides an excellent 3D image, easy to use in surgical routine, and has a satisfactory operative field depth (reaching 6–7 cm while holding the exoscope at a distance of about 40 cm). By allowing simultaneous visualization to all team members, the instrumentation allowed greater surgical precision and the absence of major complications, and the state of surgical margins on the samples were 10/10 negative (>3 mm) [5].

#### 3.7.2. Laryngeal Surgery

Due to anatomical conditions and characteristics of pathologies, laryngeal surgery requires direct visualization of the larynx both in diagnosis and in surgical treatment of benign and malignant lesions, allowing an enlargement and clear visualization of the larynx. The most recent technological developments have proposed the exoscope as a potential substitute for operating microscopes in microlaryngeal surgery, and it has been highlighted that this system has intrinsic advantages [2,23,24]. The set of features (such as depth of focus, ergonomics, and 3D image quality) guaranteed by the VITOM 3D exoscope are particularly suitable for microlaryngeal surgery (M.L.S.). Carlucci et al. (2012) reported the case of 12 patients with benign and malignant pathologies of the vocal cords treated with an exoscope [23]. 

The VITOM 3D telescope was positioned 25 cm from the operating field in place of the operating microscope behind the patient’s head. The authors emphasized the excellent sense of depth and light transmission, providing an enlarged view of the vocal cords and details of lesions such as fine vascularization, nodules, and irregularities of the mucosa. A total of 12 patients were treated: 4 with Reinke’s edema, 2 with Reinke’s space cysts, 3 with vocal cord polyps, and 1 sulcus with double chordotomy. In all patients, the VITOM 3D system allowed an excellent surgical result, stable at the subsequent postoperative checks of the laryngoscopic voice and in narrow band imaging (N.B.I).

De Virgilio et al. recently conducted a prospective pilot study to evaluate the feasibility of “exolaryngoscopic” surgery using the VITOM 3D exoscope and ARTip ™ robotic cruise system [2]. Surgeons with experience in the technology were enrolled and completed a 4-point Likert-scale questionnaire after each simulation, to rate various elements such as image quality, stereoscopic effect for surgery, magnification rate, maneuverability, and natural posture. Although promising outcomes were reported as a level of satisfaction with the device in up to 90% of cases in the presence of surgeons experienced in laryngeal surgery with traditional methods, the study design has a fundamental limitation due to the lack of comparison with procedures performed using conventional techniques. 

The feasibility of the exoscopic system has been applied in both benign and malignant pathologies of the larynx [23,24]. Cantarella et al. (2021) analyzed the results of phono surgery with this innovative system during six consecutive procedures, including two polyps, two cysts, Reinke’s edema, and unilateral paralysis of the vocal cords fat-augmented [24]. All the procedures performed demonstrated at follow-up a significant improvement of glottic closure and perceptual assessment through the VHI-10 and GIRBAS scale (*p* < 0.05 for all) at 30 days after phonosurgery.

Further technological advancements allowed the three-dimensional technology application to be added to the use of the CO_2_ laser, which had initially presented some application difficulties to laryngeal surgery. The use of the VITOM 3D system coupled with a support arm with a laser micromanipulator was recently analyzed in a preliminary study [25]. The authors reported a cohort of 17 patients with pharyngolaryngeal neoplasms treated surgically with 7 cordectomies, 2 partial supraglottic laryngectomies, 4 tongue base resections, and 4 lateral oropharyngectomies or hypopharyngectomies. The procedure achieved a low rate of deep (6%) or superficial (12%) positive margins, proving to be a safe and reliable platform, comparable to the outcomes obtained in the control group (*p* > 0.05).

#### 3.7.3. Parotid Gland Surgery

The VITOM 3D system was used to carry out a detailed surgical dissection of the parotid area, with a 3D magnification of the anatomical structures and a lower threat of iatrogenic damage to the facial nerve. 

Bartkowiak et al. conducted a prospective study on patients with benign parotid gland tumors indicated for surgical resection, comparing the surgical results of the VITOM 3D system (n = 31) with the traditional system with an operating microscope (n = 40). The authors randomized 71 patients to traditional or VITOM-assisted surgery and analyzed outcomes such as visualization quality (major auricular nerve, digastric muscle, tragal pointer), operative time, conversion rates, and surgical outcomes. Although no significant differences were found between the two approaches regarding duration of surgery (97.9 ± 40.8 min vs. 92.1 ± 39.8; *p* = 0.551), superficial parotidectomy performed (n = 10; 33.3% vs. n = 8; 34.7% *p* = 0.938), or bleeding (n = 4; 10% vs. 4; 12.9%, *p* = 0.701), and no wound revision occurred in either group, a significantly higher rate of subjects in the exoscope group developed temporary facial nerve paralysis (n = 9; 29% vs. n = 4; 10%) [4]. Therefore, while these results on visualization for parotid gland surgery with the VITOM 3D system are promising, further evidence is needed to ensure its efficacy compared to the operating microscope. Mincione et al. (2021) illustrated their background in a retrospective analysis of nine parotidectomies with benign diseases [26]. Eight tumors were located in the superficial lobe, while one was deep. The authors performed a superficial type II parotidectomy (according to the ESGS classification) in five cases (55.6%): type I–II in two cases (22.2%), type I and III in one case (11.1%). The postoperative course was good for all subjects without facial disorders. The mean operating time was 145 min (range 135–165 min). Asthenopia never occurred, and there were no cases in which the first surgeon, the assistants, or the nurses needed to interrupt the 3D vision [26]. Carta et al. (2020) reported their experience regarding parotidectomy performed under a three-dimensional, high-definition exoscope with the aim of evaluating its effectiveness in parotid surgery [27]. All nine patients underwent parotidectomy for extra facial primary tumors without preoperative involvement of the skin or of the facial nerve from March 2019 to June 2019, with the use of a 3D-HD exoscope. They did not have patients with postoperative complications or definitive facial palsy. Although the authors reported promising data on the 3D exoscope related to iatrogenic lesions of the facial nerve, the actual advantage of the method compared to traditional techniques was not adequately assessed; they did not select a control group with homogeneous patients in terms of characteristics, and the sample sizes were not adequate.

#### 3.7.4. Submandibular Gland Surgery 

The application of the 3D exoscope has found widespread use in salivary stone surgery. Previous studies found this approach to be effective, quick, and safe for the removal of large immobile stones from the hilum of the submandibular gland to the papilla. The literature shows a success rate of 87–99%, with symptom relief in 76–96% of patients. A 0–4% risk of permanent lingual nerve damage was demonstrated, and there was a low infection rate of 0–10% [28]. The confined workspace in this type of surgery is often difficult to share with not only the assistant surgeon, but also with trainees or graduates. The VITOM 3D system can provide an adequate magnification of this anatomical site and a good possibility of teaching by giving the same vision as the first surgeon to all the medical staff present in the operating room. A 24-year-old woman diagnosed with distal Wharton’s duct sialolithiasis underwent transoral removal under local anesthesia through the 4K 3D exoscope. Thanks to a high-quality enlargement of the oral pelvis, it was possible to easily identify the duct entrance of the left submandibular gland and the calculus. Wharton’s duct was engraved and the stone was removed. No postoperative complications were reported, and at 7 days of postoperative follow-up, the patient had developed a neo-ostium 5 mm from the papilla. To perform a transoral removal, the stone must be palpable, and there must be no major inflammation in the area. Since the lingual nerve usually runs under Wharton’s duct, a large palpable stone would provide protection for the nerve. If the dissection of the duct is done carefully to expose it, a small incision can be made. Several authors have advocated systematic dissection of the lingual nerve, but Ferreli et al. were opposed to this if the stone is clearly visible and has no adhesion to surrounding tissues [29]. The procedure was chosen for the first time, as it was considered the safest method of avoiding potential damage to the lingual nerve, considering the position of the stone. In fact, stones located within the anterior 1.5 cm of Wharton’s duct usually do not correspond to any risk of damage to the lingual nerve. The clinical picture and C.T. imaging were highly suggestive of a single distal stone, which is why sialoendoscopy of the submandibular duct was not performed. The exoscope can be even more beneficial in situations, such as when the stone is more posterior or when it is partially submerged below the mylohyoid muscle, with a greater risk of surgical damage. Exoscopic lingual nerve dissection should be considered if the preoperative ultrasound shows a large stone located posterior to or below the mylohyoid muscle. The exoscope would thus represent an excellent alternative for transoral excision of stones, allowing for careful surgical dissection, thus reducing the risk of iatrogenic injury. 

### 3.8. Reconstructive Approaches 

The exoscope can be considered to perform reconstructive head and neck free flap techniques, mostly during the execution of microvascular anastomoses. De Virgilio et al. (2020) studied the experience with the VITOM 3D system in a human clinical study of free flap harvesting and microvascular anastomosis in patients undergoing reconstruction after ablative surgery for head and neck carcinoma [30]. They performed 10 arterial and venous anastomoses, without significant complications. The study demonstrates the valuable alternative of the exoscope in place of the operative microscope. In a recent study, De Virgilio et al. demonstrated the use of an 3D exoscope to perform transoral excision of a tumor of the soft palate, free flap harvesting and its insetting to reconstruct the intraoral defect, and microvascular anastomosis [31]. During reconstructive surgery, the intraoral insetting of a flap is one of the most uncomfortable surgical steps for the first and second surgeons, as it forces them to work with non-ergonomic postures. With the use of the VITOM 3D system, surgeons could maintain a comfortable position, and all the observers in the operating room could share a clear and magnified view of the intraoral surgical field with the first surgeon. Moreover, the exoscope allows precise placement of sutures, which is fundamental for the reconstruction of oral cavity defects in order to reduce the risk of suture dehiscence and consequent salivary fistula [30,31].

Belykh et al. (2018) evaluated microvascular approaches via 3D exoscope or endoscope visualization [32]. Using 3D exoscopic visualization, six consecutive end-to-side microvascular anastomoses were completed on rat carotid arteries over four consecutive practice sessions. Patency of anastomoses was confirmed by indocyanine green injection in 5/6 anastomoses. Depth perception at high magnification (10–15×) was sufficient to perform delicate microsurgical manipulations such as puncturing a vessel wall and knot tying. Although the performance of anastomoses was feasible, subjective observation appraisal was that standard microscopic optics and visualization through eyepieces provided slightly better (i.e., wider field) perception of tissues at various depths, which was attributed to the ability of physiologic eye accommodation. 

The rounded area scene via the oculars was bigger than the complementary rectangular one on the monitor. Nevertheless, the highest magnification permitted small-caliber vessel details (adventitia or intima) to be more distinctly and sharply visualized via eyepieces than a 3D 4K resolution monitor.

### 3.9. Study Limitations

Although promising, the literature exhibits several limitations, including study design, sample enrollment, and few outcomes adequately investigated. The surgical time, for instance, demonstrated debated outcomes, indicating that the exoscope system may be less compliant in certain surgical areas. From this viewpoint, further scientific proof is required to clarify whether benefits such as two-hand surgery could effectively lower surgical time and demonstrate the exoscopic procedure’s advantageous cost benefit. Indeed, it was appraised that such system features could be limited if a deeper surgical field or poor lighting due to small surgical corridors are needed. Although one paper reported that the system is more efficient and cheaper than the latest operating microscope systems, other evidence suggests a deterioration of the surgical images, with consequent pixilation and excessive visual fatigue due to the use of polarizing glasses, thus questioning the economic benefit. Moreover, it should be noted that although the results shown in different studies are promising and do not demonstrate significant differences between the control groups with the traditional surgical method, the data must be interpreted cautiously, as the number of patients enrolled is often insufficient.

## 4. Conclusions

In this preliminary study, we show the great potential of this new technology, highlighting the advantages of the easy involvement of the surgical team and observers. In recent years, E.N.T. surgery has benefitted from new technologies and techniques. An increasing number of surgeons are using this technology, gaining scientific relevance every year. On the other hand, the VITOM 3D system has some limitations that can probably be overcome by advancing technology in the near future. However, today, these advantages are not enough to abandon the operating microscope. Overall, we believe that the improvements in this field of application are exponential, certainly leading to great results in the future.

## Figures and Tables

**Figure 1 jcm-11-03639-f001:**
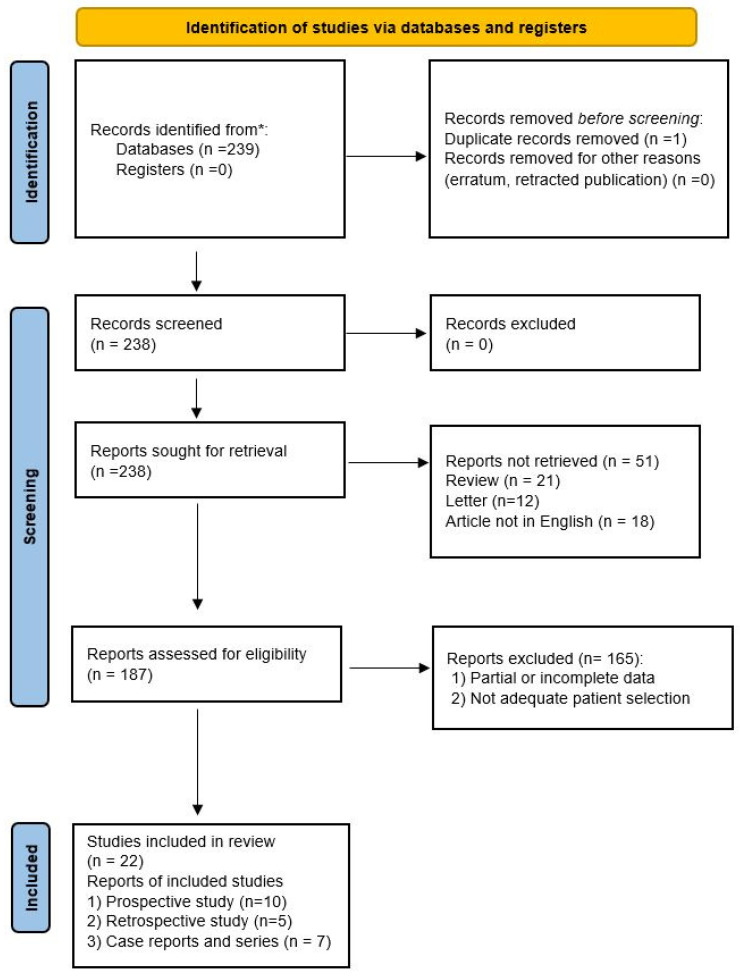
PRISMA flow diagram explaining studies’ selection process. * PubMed, Scopus, and Web of Science electronic databases.

**Figure 2 jcm-11-03639-f002:**
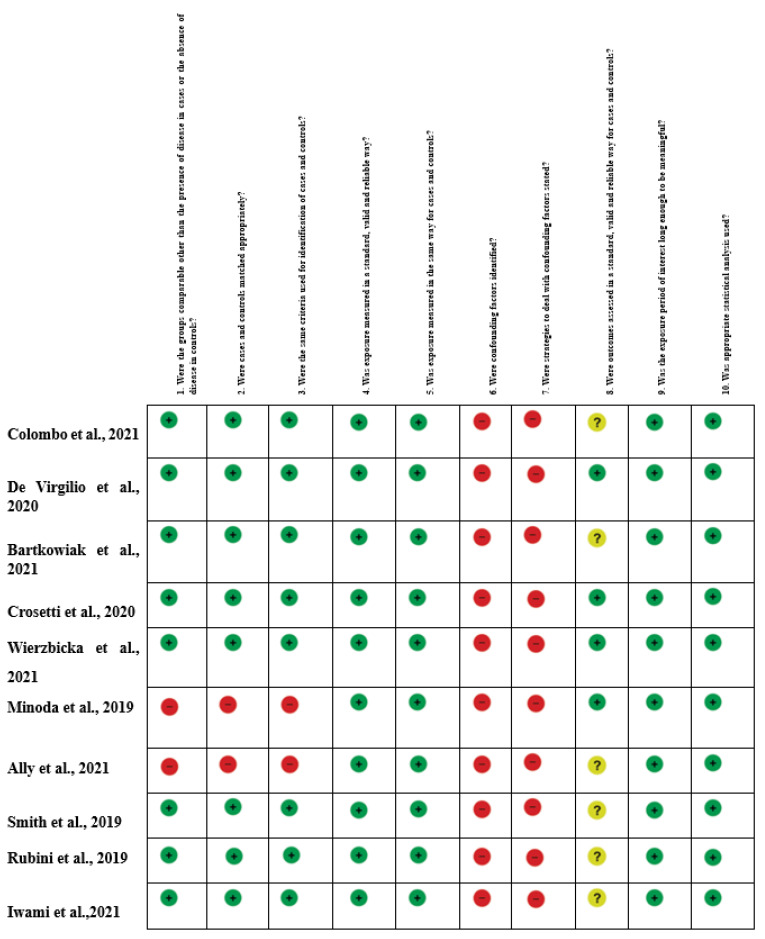
Risk of Bias summary of authors’ judgments for each study, assessed by the Joanna Briggs Institute (J.B.I.) Critical Appraisal Checklist for Case-Control studies [1,2,4,5,14,15,16,17,18,19,20,21,22,23,24,25,26,27,28,29,30,31].

**Figure 3 jcm-11-03639-f003:**
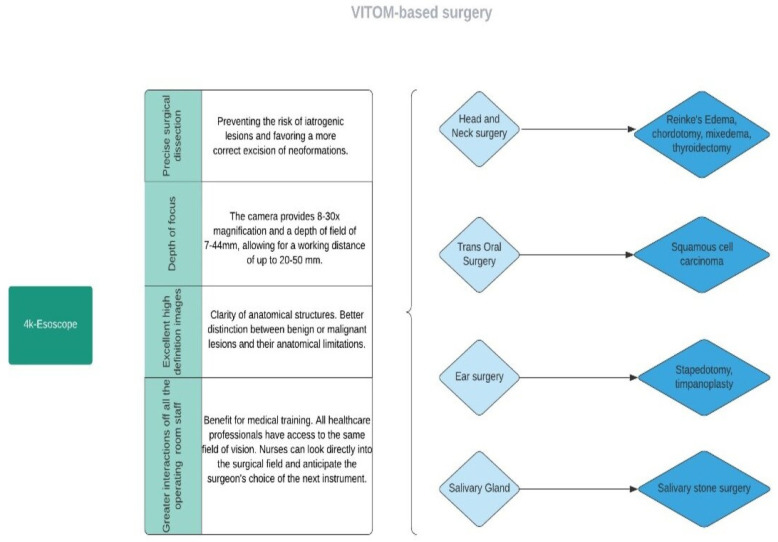
Flow diagram representing the main applications of exoscope-based surgery.

**Table 1 jcm-11-03639-t001:** Studies retrieved in qualitative analysis, outcomes comparison and cost effectiveness.

Authors, Year, Reference	Study Design	Patients	Treatment	Results	Surgical Techniques: Comparison and Complications	Surgical Times, Cost Effectiveness
Colombo et al., 2021 [1]	Retrospective study	13	Ear surgery	There was no significant statistical difference between the two groups concerning ORst1 (*p* = 0.90), ORst2 (*p* = 0.76), and operation time (*p* = 0.59). The tympanoplasty with mastoidectomy median operation time was 163 min (IQ25–75: 144–210) in the exoscope group and 148 min (IQ25–75: 128–165) in the operative microscope group; no statistical difference was found between the two groups (*p* = 0.23). The operation time for binaural cochlear implant case number 12 was 152 min using the exoscope and 158 min using the operative microscope; for case number 13, the operation time was 174 min with the exoscope and 211 min with the operative microscope.	Easier positioning of the exoscope inside the operating room due to its weight being significantly lower than that of the operative microscope (0.7 kg vs. > 229 kg). It is easier to maneuver, and it occupies a smaller amount of space. Advantages include lightness, maneuverability, and compactness. It can be easily one-handedly rotated and moved in any direction, achieving even narrow view angles. The exoscope needs a large surgical corridor to guarantee good performance. The wider variety of visual angles in exoscope surgery makes it necessary to define a new anatomical point of view, with specific training.	The exoscope was cheaper than a modern operative microscope. No significant differences were reported in the surgical times in group comparison (*p* < 0.05).
De Virgilio et al., 2021 [2]	Prospective study	10	Micro-Laryngeal Surgery	The highest mean placement time in medical students and residents (81.29 s and 57.96 s, respectively) compared to E.N.T. staff members (45.10 s). Furthermore, among the outcomes, a level of satisfaction with the method was recorded in up to 90% of cases.	Maneuvering is intuitive thanks to manual movements via direct drag mode. Setup of the system is easy and fast thanks to the position preset, which memorizes standard robotic arm positions. It is, in fact, possible to save and recall previous positions without directly touching the robotic arm.	The mean times of the procedures were similar between the three groups (*p* < 0.05 for all).
Bartkowiak et al., 2021 [4]	Prospective study	71	Parotid surgery	A significantly higher percentage of patients in the exoscope group developed transient facial nerve paralysis (*n* = 9; 29% vs. *n* = 4, 10%).	A higher degree of movement freedom and excellent ergonomics with reduced fatigue. No exoscope-related complications were observed.	Both microscope and exoscope procedures had similar total operative times (>90 min).
Crosetti et al., 2020 [5]	Prospective study	10	Oropharyngeal surgery	Post-surgical pain was low; NRS: 1.2. One case of subcutaneous emphysema and one submental blood collection.	VITOM 3D proved to be a versatile and compact optical instrument, giving an excellent 3D image without oral cavity involvement. Only 2/41 ORL staff developed discomfort due to the 3D vision: headache and pain in the nose bridge.	The average cost of consumables (VITOM and joystick sterile coating) per procedure was €62 (€41 and €21, respectively). Effectiveness is comparable with TORS and microscopic transoral techniques, but with lower platform costs.
Wierzbicka et al., 2021 [14]	Prospective study	60	Otosurgery	The differences between the 3D exoscope and the microscope were not statistically significant (*p* = 0.488).	High-resolution 3D images, greater freedom for exoscope adjustment, and a comfortable surgical posture. In deeper areas of the middle ear, due to the surgical field narrowing, the exoscope provided worse visibility.	The average time for the procedure did not differ from the microscope for stapes (40 min). Contrasting field visualization depending on surgical steps.
Minoda et al., 2019 [15]	Case series	2	Middle ear cholesteatoma surgery	No residual cholesteatoma after 9 months postoperatively.	The higher magnification using the system caused a deterioration of the surgical images. Uncomfortable refocusing of the surgical 3D exoscope system.	It was quick and smooth, unlike the transition between microscope and endoscope.
Ally et al., 2021 [16]	Case report	1	Mastoid surgery	No postoperative complications. The patient was discharged on day 1 postoperatively because of comorbidities. At the 2-week follow-up, the graft had taken well, and there was no evidence of any remnant disease.	Head-up position procedure, more comfortable. Eight times the depth of field compared to the microscope and nearly twice the magnification. The light was too bright down the external auditory canal through a speculum.	The cost of the exoscopic platform (approximately £120,000) is comparable to the operating microscope and about 10 times lower than the da Vinci robotic surgical system.
Smith et al., 2019 [17]	Prospective study	11	Lateral skull base surgery	The exoscope was the sole visualization tool in 7 cases, with 4 including the use of an endoscope or microscope. There were no perioperative complications. Potential subjective advantages include superior ergonomics, compact size, and an equal visual experience for surgeons and observers.	New visualization system has a learning curve. Low lighting in small surgical corridors and pixilation at high magnification.	Surgeons became subjectively more comfortable and efficient with repeated exoscope usage.
Rubini et al., 2019 [18]	Retrospective study	24	Lateral skull base surgery	No intraoperative complication, while postoperatively, only one minor complication emerged. The facial and hearing function outcomes were fully comparable (*p* > 0.05).	The anatomical structures are more realistic, and the recognition and differentiation of the different structures are better with the 3D exoscopic view. The exoscope allowed a position with a horizontal view throughout the surgical operation. Surgeon experiences discomfort due to the use of fixed optics and limited movements.	The average operative time was 289 min in the exoscope group and 313 min in the microscope group.
Iwami et al., 2021 [19]	Retrospective study	8	Craniofacial Resection	The exoscope provided excellent 3D transcranial images in all the patients treated. The exoscopic transcranial approach was performed according to the same steps as the traditional C.F.R., which used a microscope.	Ability to be conveniently and rapidly interchanged with an endoscope improved ergonomics and shared operative view. Limited illumination and pixilation at high magnifications when using deep and narrow surgical corridors.	Lower costs, ease of portability, and educational benefits.
Tasca I. et al., 2016 [20]	Case report	1	Rhinoplasty	Visualization during interventions was improved, thereby allowing understanding of the procedures and enhancing the teaching environment.	High-definition images, whilst simultaneously maintaining the use of two operating hands. The operating field may always be centered on the screen even in cases of inevitable movements of anatomical structures during operating maneuvers, such as elevations of the tunnels or osteotomies.	Although several advantages were reported, the authors stated the system could not represent a substitute for traditional surgery.
Pirola et al., 2021 [21]	Retrospective study	21	Lacrimal surgery	At follow-up, 1/31 (3.2%) cases had persisting unilateral epiphora in the exo-endoscopic group, with 2/53 (3.8%) in the endoscopy-only group. No statistical differences among unsuccessful procedures (*p* = 0.896).	The concurrent picture in picture visualization has educational potential. Reduced stereoscopic effect, uncomfortable position, and increased eye strain due to watching a screen.	The mean procedure times were similar among the three groups enrolled (20.22 s, 21.92 s, and 22.59 s).
Bignami et al., 2021 [22]	Case report	1	Frontal Fibro-osseous removal	Effective removal of the lesion and good access to the whole frontal sinus, with proper control of critical areas. No complications.	High-quality recordings that are extremely useful for didactic purposes. Anatomical structures were enhanced and magnified for better appreciation. Additionally, the perspective is always the same as the main surgeon’s, which is an additional factor that might facilitate the transition from the role of assistant to first operator. “Head-up surgery” might be unfamiliar or difficult	The authors reported reduced operative time with consequent cost-effectiveness benefits.
Carlucci et al., 2012 [23]	Prospective study	12	Laryngeal surgery	Postoperative voice analysis showed a good result in the resolution of phonatory problems. Laryngeal biopsies were easy to obtain, as was the use of N.B.I.	Excellent visualization of laryngeal structures (especially Reinke’s edema), chordal cysts, N.B.I. usage, and proposed biopsies for neoformation. Limited sulcus visualization with the exoscope system.	-
Cantarella et al., 2021 [24]	Case series	6	Phonosurgery	Significant outcomes in dysphonia (*p* = 0.03), VHI-10 (*p* = 0.03), voice breathiness (*p* = 0.03), and maximal phonation time (*p* = 0.03).	Optimal depth of field. High illumination and definition of anatomical detail. The monitor needed to be oriented perpendicular to the surgeon’s view. No technical difficulty.	The time required to set up the equipment and perform the procedures was similar to the operative microscope.
Carobbio et al., 2021 [25]	Prospective study	17	Transoral microsurgery	Surgical times for both laryngeal and oropharyngeal/hypopharyngeal lesions (*p* = 0.17 and *p* = 0.59, respectively).	The smaller size of the viewing system allows for better ergonomics for both the first surgeon, the assistant, and the entire surgical team, improving the assistant’s maneuvers and teaching purposes. The 3D-HD exoscope has minimal image latency, especially observed during hand movements at high magnification.	Time-sensitive oncologic procedures such as transoral laser microsurgery or transoral resection of critical laryngotracheal stenosis could be carried out with the esoscopic system.
Mincione et al., 2021 [26]	Retrospective study	9	Parotid gland surgery	Superficial parotidectomy was performed in 5 cases (55.6%): type I–II in 2 cases (22.2%), type I and III in 1 case (11.1%). The postoperative period was uneventful for all patients, and no complications were reported. The mean operating time was 145 min (range 135–165 min).	Visualization, ergonomics, versatility, training, and education. Asthenopia and a long learning curve were reported.	-
Carta et al., 2020 [27]	Prospective study	9	Parotidectomy	No statistical significance in the range of postoperative transient facial nerve weakness of the present series (11.1%) and the range of the previous study (5.9%) (*p* = 0.532) in comparison.	The implementation of lighting and magnification of the surgical field and the capability for precise dissection of fine structures. Occurrence of fatigue, headache, dizziness, and eye strain during or after surgery due to the polarizing glasses needed to provide a 3D view for the length of the procedure.	The mean (range) time of surgery was 149.4 (115–210) min. No postoperative complications were experienced; the mean hospitalization time was 3.8 days.
Capaccio et al., 2011 [28]	Prospective study	5	Submandibular gland surgery	Successful stone removal and significant subjective improvement were achieved. One of the patients revealed a residual 3 mm asymptomatic Hilo-parenchymal stone, consequently removed. Wharton’s duct and the lingual nerve were identified and preserved in all cases. Only two patients required sialendoscopy to locate the stone more precisely.	A better view of the operating field by all members of the surgical team; clear anatomical delineation; and improved oral floor depth perception, lingual nerve, and Wharton’s duct.	No substantial difference between 2D and 3D endoscopic surgery in terms of blood loss and operative times.
Ferreli et al., 2020 [29]	Case report	1	Transoral surgery of calculus of Wharton’s duct	High-quality magnification of the oral pelvis, easy identification of the entrance of the left submandibular gland duct, and the calculus was obtained. No postoperative complications occurred. At 7 days postoperative follow-up, the patient had developed a neo-ostium 5 mm from the papilla.	The risk for iatrogenic lesions of the lingual nerve was reduced. However, it was possible to treat only palpable and anterior stones.	-
De Virgilio et al., 2020 [30]	Prospective study	10	Free flap head and neck reconstruction	The exoscope system provided sufficient access, reach, and visualization to perform successfully free flap harvesting and microvascular anastomosis. No significant complications.	Using the 3D glasses, both surgeons and nurses benefit from the same visualization throughout the entire procedure	-
De Virgilio et al., 2021 [31]	Case report	1	Soft palate excision and reconstruction	Exoscopic technology has been proven to benefit each step of head and neck demolition and reconstruction. The magnified surgical field obtained with the exoscopic technology permits higher surgical precision.	The necessity to wear 3D glasses can be bothersome for some operators because the lenses are dark, and the exoscope is used in half-light to enhance the visual quality of the screen.	The authors suggested conducting more specific studies to define the potential impact of the exoscope on setup and surgery time.

## Data Availability

Not applicable.

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
