# Peer review of "High Definition Three-Dimensional Exoscope (VITOM 3D) in E.N.T. Surgery: A Systematic Review of Current Experience"

_jcm, 2022, doi:10.3390/jcm11133639_

Round 1

Reviewer 1 Report

The topic of this paper is fresh in the head & neck surgery. The study is well designed and written. The references are up to date.

I found two problems:
1. In my opinion table 1 is to large and detailed. Data retrieved from original studies are discussed again in the text. Is is possible to make it more compact?

2. The references No 1/15 and 4/27 were duplicated. The references should be formatted accortdin rules of the journal.

Author Response

Response point by point to the reviewers

Dear editor thanks for the suggestions to improve the quality of the paper.

Reviewer #1: The topic of this paper is fresh in the head & neck surgery. The study is well designed and written. The references are up to date.

  1. Comments: In my opinion table 1 is to large and detailed. Data retrieved from original studies are discussed again in the text. Is possible to make it more compact?

Response: Dear revisor, thanks for the suggestions. We tried to make more compact the table but another revisors expressed the need of improve some aspects of the topic thus we could not reduce more.

  1. Comments: The references No 1/15 and 4/27 were duplicated. The references should be formatted according to rules of the journal.

Response: Dear reviewer thanks for the suggestions. We’ve corrected the duplicate and references order.

Best regards.

Reviewer 2 Report

The newly developed exoscope system provides high-definition images and accuracy magnification during operation. The authors made a systemic review of 239 potentially relevant studies in this article. They enrolled 23 papers for the final data analysis to evaluate the application of VITOM 3D exoscope in the surgical field of ENT.

There are some weak points in the manuscript.

  1. Although the authors use the systemic review analysis, there are no consistent definite outcomes of all the enrolled papers. The demonstrated results are heterogeneous outcomes in Table I. The authors should provide the major compared surgical outcomes, such as surgical times, cost, or complications, between similar surgery.
  2. Authors did not demonstrate apparent benefits of the VITOM 3D system compared to traditional tools. The authors just described the application of VITOM 3D systems in different ENT fields.
  3. The resolution of Figure 2 is poor for reading.
  4. The specific abbreviation of VITOM 3D is inconsistent in the all manuscript, for example, VITOM -3D, Vitom 3D, VITOM 4K 3D
  5. Please unify all the manuscript's repeat terms and the corresponding abbreviation (O.M, HD, EX, DCR, OPF, 3Des).

Author Response

Response point by point to the reviewers

Dear editor thanks for the suggestions to improve the quality of the paper.

Reviewer #2: There are some weak points in the manuscript.

Comments: Although the authors use the systemic review analysis, there are no consistent definite outcomes of all the enrolled papers. The demonstrated results are heterogeneous outcomes in Table I. The authors should provide the major compared surgical outcomes, such as surgical times, cost, or complications, between similar surgery.

Response: dear revisor, thanks for the suggestions. We’ve remodulated the table according to the suggestions, specifying into different sections the main topics suggested.

Comments: Authors did not demonstrate apparent benefits of the VITOM 3D system compared to traditional tools. The authors just described the application of VITOM 3D systems in different ENT fields.

Response: dear revisor, thanks for the interesting and correct suggestion. We’ve reanalyzed all the papers included and although reduced data in regard, we wrote and added to the results a specific section tryng to expleain all the topica s surgical times, advantages and disadvantages of the procedure compared to the traditional one and cost-benefit. In this last topic there were poor data but we argumented as possible.

We’ve added also in study limitations all the main topics that should be discussed in the future

Comments: The resolution of Figure 2 is poor for reading.

Response: Thanks for the suggestions. We improved to 300 dpi the quality of the images. However, due to the format of the tool, it’s not possible to increase more the size of the sentences.

Comments: The specific abbreviation of VITOM 3D is inconsistent in the all manuscript, for example, VITOM -3D, Vitom 3D, VITOM 4K 3D

Response: Dear reviewer we agree with the suggestions. We’ve modified all the terms present in the paper.

Comments: Please unify all the manuscript's repeat terms and the corresponding abbreviation (O.M, HD, EX, DCR, OPF, 3Des).

Response: Dear reviewer, we have corrected all abbreviations, removed where incorrect or corrected and brought into line with the consequent ones as suggested.

Best regards.

Reviewer 3 Report

The meta-analyze is created good but the part focused to thyroid surgery based on one case report only. I am sure that this part is not be able for meta-analyze generally.

The meta-analyze must be based on the collection of strong data. The using of one case for a part of analyze of thyroid surgery must be improved by supporting by more cases of thyroid surgery patients large number of them or must be delete from a text. The same problem exist for more parts than the analyze based on the small number of operation. The authors described the meta-analyze on the first data in literature that they can be first with this theses.

Author Response

Response point by point to the reviewers

Dear editor thanks for the suggestions to improve the quality of the paper.

Reviewer #3:

Comments: The meta-analyze is created good but the part focused to thyroid surgery based on one case report only. I am sure that this part is not be able for meta-analyze generally. The meta-analyze must be based on the collection of strong data. The using of one case for a part of analyze of thyroid surgery must be improved by supporting by more cases of thyroid surgery patients large number of them or must be delete from a text. The same problem exist for more parts than the analyze based on the small number of operation. The authors described the meta-analyze on the first data in literature that they can be first with this theses.

Response: Dear reviewer, thanks for the suggestions. We agree with what you have stated. Since the topic is innovative and the evidence in the literature is limited to date, we have included all possible data, albeit rightfully for the meta-analysis process. We have therefore removed the paragraph on thyroid surgery as only a case report is available. We removed also and modified the prisma diagram, the references and the pooled data of the results. The others evidence included such as case reports were instead considered valid since added to other studies with larger samples and therefore an adequate total number was reached in the pooled analysis.

Thanks for the work done.

Best regards.

Round 2

Reviewer 2 Report

In this article, the authors try to compare the applications of 3D exoscope with traditional surgical tools. The content is more like the general introduction of current applications of VITOM 3D in ENT surgery. However, the results demonstrated the inconsistent and heterogeneous surgical benefits due to different operative fields. In addition, the cited content is not appropriate, with almost similar sentences to the original articles. (Page 18, Line 250-262; Page 23, line 363-376; Page 23, Line 377-395; Page 26, Line 453-465).

Author Response

Dear Reviewer,

thanks for the suggestions and indications, during the revisions we already started the changes to reduce the plagiarism as indicated also by the editor. Unfortunately frequently the paper could report several repeated word or sentences due to the tecnical language and we hope that all the changes required were made to improve the readability. Below all the sentences modified.

Best regards:

- Surgical time, facial and hearing function outcomes, and intraoperative or postoperative complications were analyzed, demonstrating no intraoperative complications occurred during all the procedures. In contrast, at follow-up, one complication occurred. The authors reported an average surgical time of 289min in the exoscope group while 313 min in the microscope one, not revealing statistical differences (p>0.05). Moreover, the facial and hearing function outcomes were similar [15].

3.4 Nasal and paranasal surgery

Nasal and paranasal surgery could benefit from exoscopic and 3D techniques as the outstanding visualization of the anatomical structures is an essential element for surgical success [16,17]. Optical magnification keeps an essential role in rhinology techniques, particularly after the spread of endoscopic approaches for nasal surgery.

  • No exoscope-related difficulties occurred. Five patients experienced an exoscope-assisted profound lobe approach that needed intraoperative conversion to the microscope technique. No discrepancies were found in the subjective grade of intraoperative visualization of critical anatomical structures. However, a significantly greater rate of subjects in the exoscope group developed temporary facial nerve paralysis (n = 9; 29% vs. n = 4; 10%). Although these results indicate that the VITOM 3D is a useful visualization instrument for parotid gland surgery, more evidence is required to ensure its effectiveness [4].

Mincione et al. in 2021 illustrate their background in a retrospective analysis of 9 parotidectomies cases with benign diseases [25]. Eight tumors were located in the superficial lobe, while one was deep. The authors performed a superficial parotidectomy type II (according to the ESGS classification) in 5 cases (55.6%): type I-II in 2 cases (22.2%), type I and III in 1 case, respectively (11.1%). The postoperative course was good for all the subjects with no facial disorders.

  • The rounded area scenery via the oculars was bigger than the complementary rectangular one on the monitor. Nevertheless, the highest magnification permitted small-caliber vessel details (adventitia or intima) to be more distinctly and sharply visualized via eyepieces than a 3D 4K resolution monitor.

3.9 Studies Limitations

Although pledging, the data currently in the literature contains several limitations, including study design, sample enrollment, and few outcomes adequately investigated. The surgical time, for instance, demonstrated debated outcomes, indicating that the esoscope system may be less compliant in certain surgical areas. In this viewpoint, further scientific proof is required to clarify whether benefits such as 2-hand surgery could effectively lower surgical time and demonstrate the exoscopic procedure's advantageous cost-benefit. Indeed, it was apprised that such system features could be limited if a deeper surgical field or poor lighting due to small surgical corridors are needed.

The references were also checked for eligibility and no self-citations is present.

Best regards.
